# Discovering, Learning and Exploiting Relevance

**Cem Tekin**
Electrical Engineering Department
University of California Los Angeles
cmtkn@ucla.edu

**Mihaela van der Schaar**
Electrical Engineering Department
University of California Los Angeles
mihaela@ee.ucla.edu

## Abstract

In this paper we consider the problem of learning online what is the information to consider when making sequential decisions. We formalize this as a contextual multi-armed bandit problem where a high dimensional ($D$-dimensional) context vector arrives to a learner which needs select an action to maximize its expected reward at each time step. Each dimension of the context vector is called a *type*. We assume that there exists an unknown relation between actions and types, called the *relevance relation*, such that the reward of an action only depends on the contexts of the relevant types. When the relation is a function, i.e., the reward of an action only depends on the context of a single type, and the expected reward of an action is Lipschitz continuous in the context of its *relevant* type, we propose an algorithm that achieves $\tilde{O}(T^\gamma)$ regret with a high probability, where $\gamma = 2/(1 + \sqrt{2})$. Our algorithm achieves this by learning the *unknown* relevance relation, whereas prior contextual bandit algorithms that do not exploit the existence of a relevance relation will have $\tilde{O}(T^{(D+1)/(D+2)})$ regret. Our algorithm alternates between exploring and exploiting, it does not require reward observations in exploitations, and it guarantees with a high probability that actions with suboptimality greater than $\epsilon$ are never selected in exploitations. Our proposed method can be applied to a variety of learning applications including medical diagnosis, recommender systems, popularity prediction from social networks, network security etc., where at each instance of time vast amounts of different types of information are available to the decision maker, but the effect of an action depends only on a single type.

## 1 Introduction

In numerous learning problems the decision maker is provided with vast amounts of different types of information which it can utilize to learn how to select actions that lead to high rewards. The value of each type of information can be regarded as the context on which the learner acts, hence all the information can be encoded in a context vector. We focus on problems where this context vector is high dimensional but the reward of an action only depends on a small subset of types. This dependence is given in terms of a relation between actions and types, which is called the relevance relation. For an action set $\mathcal{A}$ and a type set $\mathcal{D}$, the relevance relation is given by $\boldsymbol{\mathcal{R}} = \{\mathcal{R}(a)\}_{a \in \mathcal{A}}$, where $\mathcal{R}(a) \subset \mathcal{D}$. Expected reward of an action $a$ only depends on the values of the relevant types of contexts. Hence, for a context vector $\boldsymbol{x}$, action $a$'s expected reward is equal to $\mu(a, \boldsymbol{x}_{\mathcal{R}(a)})$, where $\boldsymbol{x}_{\mathcal{R}(a)}$ is the context vector corresponding to the types in $\mathcal{R}(a)$. Several examples of relevance relations and their effect on expected action rewards are given in Fig. 1. The problem of finding the relevance relation is important especially when $\max_{a \in \mathcal{A}} |\mathcal{R}(a)| << |\mathcal{D}|$.[1] In this paper we consider the case when the relevance relation is a function, i.e., $|\mathcal{R}(a)| = 1$, for all $a \in \mathcal{A}$, which is an important special case. We discuss the extension of our framework to the more general case in Section 3.3.

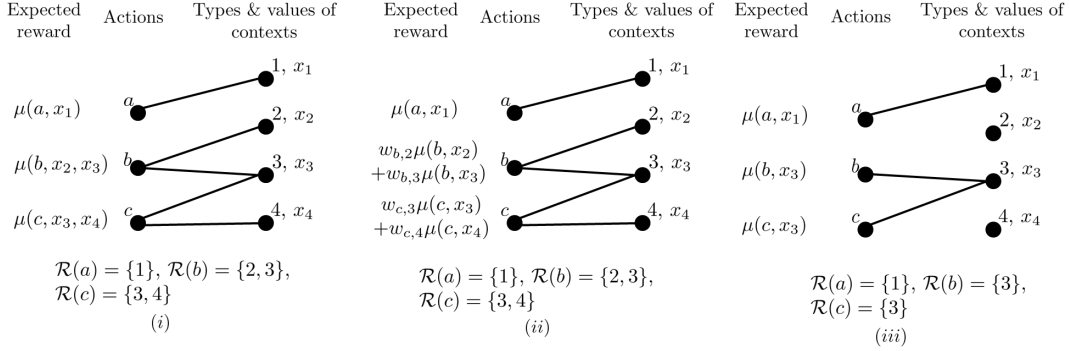

Figure 1: Examples of relevance relations: $(i)$ general relevance relation, $(ii)$ linear relevance relation, $(iii)$ relevance function. In this paper we only consider $(iii)$, while our methods can easily be generalized to $(i)$ and $(ii)$.

Relevance relations exists naturally in many practical applications. For example, when sequentially treating patients with a particular disease, many types of information (contexts) are usually available - the patients' age, weight, blood tests, scans, medical history etc. If a drug's effect on a patient is caused by only one of the types, then learning the relevant type for the drug will result in significantly faster learning for the effectiveness of the drug for the patients.[2] Another example is recommender systems, where recommendations are made based on the high dimensional information obtained from the browsing and purchase histories of the users. A user's response to a product recommendation will depend on the user's gender, occupation, history of past purchases etc., while his/her response to other product recommendations may depend on completely different information about the user such as the age and home address.

Traditional contextual bandit solutions disregard existence of such relations, hence have regret bounds that scale exponentially with the dimension of the context vector [1, 2]. In order to solve the curse of dimensionality problem, a new approach which learns the relevance relation in an online way is required. The algorithm we propose simultaneously learns the relevance relation (when it is a function) and the action rewards by comparing sample mean rewards of each action for context pairs of different types that are calculated based on the context and reward observations so far. The only assumption we make about actions and contexts is the Lipschitz continuity of expected reward of an action in the context of its relevant type. Our main contributions can be summarized as follows:

- We propose the *Online Relevance Learning with Controlled Feedback* (ORL-CF) algorithm that alternates between exploration and exploitation phases, which achieves a regret bound of $\tilde{O}(T^\gamma)$,[3] with $\gamma = 2/(1 + \sqrt{2})$, when the relevance relation is a function.

- We derive separate bounds on the regret incurred in exploration and exploitation phases. ORL-CF only needs to observe the reward in exploration phases, hence the reward feedback is controlled. ORL-CF achieves the same time order of regret even when observing the reward has a non-zero cost.

- Given any $\delta > 0$, which is an input to ORL-CF, suboptimal actions will *never* be selected in exploitation steps with probability at least $1 - \delta$. This is very important, perhaps vital in numerous applications where the performance needs to be guaranteed, such as healthcare.

Due to the limited space, numerical results on the performance of our proposed algorithm is included in the supplementary material.

## 2 Problem Formulation

$\mathcal{A}$ is the set of actions, $D$ is the dimension of the context vector, $\mathcal{D} := \{1, 2, \ldots, D\}$ is the set of types, and $\mathcal{R} = \{\mathcal{R}(a)\}_{a \in \mathcal{A}} : \mathcal{A} \to \mathcal{D}$ is the relevance function, which maps every $a \in \mathcal{A}$ to a unique $d \in \mathcal{D}$. At each time step $t = 1, 2, \ldots$, a context vector $\boldsymbol{x}_t$ arrives to the learner. After observing $\boldsymbol{x}_t$ the learner selects an action $a \in \mathcal{A}$, which results in a random reward $r_t(a, \boldsymbol{x}_t)$. The learner may choose to observe this reward by paying cost $c_O \geq 0$. The goal of the learner is to maximize the sum of the generated rewards minus costs of observations for any time horizon $T$.

Each $\boldsymbol{x}_t$ consists of $D$ types of contexts, and can be written as $\boldsymbol{x}_t = (x_{1,t}, x_{2,t}, \ldots, x_{D,t})$ where $x_{i,t}$ is called the type $i$ context. $\mathcal{X}_i$ denotes the space of type $i$ contexts and $\mathcal{X} := \mathcal{X}_1 \times \mathcal{X}_2 \times \ldots \times \mathcal{X}_D$ denotes the space of context vectors. At any $t$, we have $x_{i,t} \in \mathcal{X}_i$ for all $i \in \mathcal{D}$. For the sake of notational simplicity we take $\mathcal{X}_i = [0, 1]$ for all $i \in \mathcal{D}$, but all our results can be generalized to the case when $\mathcal{X}_i$ is a bounded subset of the real line. For $\boldsymbol{x} = (x_1, x_2, \ldots, x_D) \in \mathcal{X}$, $r_t(a, \boldsymbol{x})$ is generated according to an i.i.d. process with distribution $F(a, x_{\mathcal{R}(a)})$ with support in $[0, 1]$ and expected value $\mu(a, x_{\mathcal{R}(a)})$.

The following assumption gives a similarity structure between the expected reward of an action and the contexts of the type that is relevant to that action.

**Assumption 1.** *For all $a \in \mathcal{A}$, $\boldsymbol{x}, \boldsymbol{x}' \in \mathcal{X}$, we have $|\mu(a, x_{\mathcal{R}(a)}) - \mu(a, x'_{\mathcal{R}(a)})| \leq L |x_{\mathcal{R}(a)} - x'_{\mathcal{R}(a)}|$, where $L > 0$ is the Lipschitz constant.*

We assume that the learner knows the $L$ given in Assumption 1. This is a natural assumption in contextual bandit problems [1, 2]. Given a context vector $\boldsymbol{x} = (x_1, x_2, \ldots, x_D)$, the optimal action is $a^*(\boldsymbol{x}) := \arg\max_{a \in \mathcal{A}} \mu(a, x_{\mathcal{R}(a)})$, but the learner does not know it since it does not know $\mathcal{R}$, $F(a, x_{\mathcal{R}(a)})$ and $\mu(a, x_{\mathcal{R}(a)})$ for $a \in \mathcal{A}$, $\boldsymbol{x} \in \mathcal{X}$ a priori. In order to assess the learner's loss due to unknowns, we compare its performance with the performance of an *oracle benchmark* which knows $a^*(\boldsymbol{x})$ for all $\boldsymbol{x} \in \mathcal{X}$. Let $\mu_t(a) := \mu(a, x_{\mathcal{R}(a),t})$. The action chosen by the learner at time $t$ is denoted by $\alpha_t$. The learner also decides whether to observe the reward or not, and this decision of the learner at time $t$ is denoted by $\beta_t \in \{0, 1\}$, where $\beta_t = 1$ implies that the learner chooses to observe the reward and $\beta_t = 0$ implies that the learner does not observe the reward. The learner's performance loss with respect to the oracle benchmark is defined as the regret, whose value at time $T$ is given by

$$R(T) := \sum_{t=1}^{T} \mu_t(a^*(\boldsymbol{x}_t)) - \sum_{t=1}^{T} (\mu_t(\alpha_t) - c_O \beta_t). \tag{1}$$

A regret that grows sublinearly in $T$, i.e., $O(T^\gamma)$, $\gamma < 1$, guarantees convergence in terms of the average reward, i.e., $R(T)/T \to 0$. We are interested in achieving sublinear growth with a rate independent of $D$.

## 3 Online Relevance Learning with Controlled Feedback

### 3.1 Description of the algorithm

In this section we propose the algorithm *Online Relevance Learning with Controlled Feedback* (ORL-CF), which learns the best action for each context vector by simultaneously learning the relevance relation, and then estimating the expected reward of each action. The feedback, i.e., reward observations, is controlled based on the past context vector arrivals, in a way that reward observations are only made for actions for which the uncertainty in the reward estimates are high for the current context vector. The controlled feedback feature allows ORL-CF to operate as an active learning algorithm. Operation of ORL-CF can be summarized as follows:

- Adaptively discretize (partition) the context space of each type to learn action rewards of similar contexts together.
- For an action, form reward estimates for pairs of intervals corresponding to pairs of types. Based on the accuracy of these estimates, either choose to explore and observe the reward or choose to exploit the best estimated action for the current context vector.
- In order to choose the best action, compare the reward estimates for pairs of intervals for which one interval belongs to type $i$, for each type $i$ and action $a$. Conclude that type $i$

is relevant to $a$ if the variation of the reward estimates does not greatly exceed the natural variation of the expected reward of action $a$ over the interval of type $i$ (calculated using Assumption 1).

---

**Online Relevance Learning with Controlled Feedback (ORL-CF):**
1: Input: $L$, $\rho$, $\delta$.
2: Initialization: $\mathcal{P}_{i,1} = \{[0,1]\}$, $i \in \mathcal{D}$. Run **Initialize**$(i, \mathcal{P}_{i,1}, 1)$, $i \in \mathcal{D}$.
3: **while** $t \geq 1$ **do**
4:     Observe $\boldsymbol{x}_t$, find $\boldsymbol{p}_t$ that $\boldsymbol{x}_t$ belongs to.
5:     Set $\mathcal{U}_t := \bigcup_{i \in \mathcal{D}} \mathcal{U}_{i,t}$, where $\mathcal{U}_{i,t}$ (given in (3)), is the set of under explored actions for type $i$.
6:     **if** $\mathcal{U}_t \neq \emptyset$ **then**
7:         (**Explore**) $\beta_t = 1$, select $\alpha_t$ randomly from $\mathcal{U}_t$, observe $r_t(\alpha_t, \boldsymbol{x}_t)$.
8:         Update pairwise sample means: for all $q \in Q_t$, given in (2).
        $\bar{r}^{\mathrm{ind}(q)}(q, \alpha_t) = (S^{\mathrm{ind}(q)}(q, \alpha_t) \bar{r}^{\mathrm{ind}(q)}(q, \alpha_t) + r_t(\alpha_t, \boldsymbol{x}_t))/(S^{\mathrm{ind}(q)}(q, \alpha_t) + 1)$.
9:         Update counters: for all $q \in Q_t$, $S^{\mathrm{ind}(q)}(q, \alpha_t) + +$.
10:     **else**
11:         (**Exploit**) $\beta_t = 0$, for each $a \in \mathcal{A}$ calculate the set of candidate relevant contexts $\mathrm{Rel}_t(a)$ given in (4).
12:         **for** $a \in \mathcal{A}$ **do**
13:             **if** $\mathrm{Rel}_t(a) = \emptyset$ **then**
14:                 Randomly select $\hat{c}_t(a)$ from $\mathcal{D}$.
15:             **else**
16:                 For each $i \in \mathrm{Rel}_t(a)$, calculate $\mathrm{Var}_t(i, a)$ given in (5).
17:                 Set $\hat{c}_t(a) = \arg\min_{i \in \mathrm{Rel}_t(a)} \mathrm{Var}_t(i, a)$.
18:             **end if**
19:             Calculate $\bar{r}_t^{\hat{c}_t(a)}(a)$ as given in (6).
20:         **end for**
21:         Select $\alpha_t = \arg\max_{a \in \mathcal{A}} \bar{r}_t^{\hat{c}_t(a)}(p_{\hat{c}_t(a),t}, a)$.
22:     **end if**
23:     **for** $i \in \mathcal{D}$ **do**
24:         $N^i(p_{i,t}) + +$.
25:         **if** $N^i(p_{i,t}) \geq 2^{\rho l(p_{i,t})}$ **then**
26:             Create two new level $l(p_{i,t}) + 1$ intervals $p, p'$ whose union gives $p_{i,t}$.
27:             $\mathcal{P}_{i,t+1} = \mathcal{P}_{i,t} \cup \{p, p'\} - \{p_{i,t}\}$.
28:             Run **Initialize**$(i, \{p, p'\}, t)$.
29:         **else**
30:             $\mathcal{P}_{i,t+1} = \mathcal{P}_{i,t}$.
31:         **end if**
32:     **end for**
33:     $t = t + 1$
34: **end while**

---

**Initialize**$(i, \mathcal{B}, t)$:
1: **for** $p \in \mathcal{B}$ **do**
2:     Set $N^i(p) = 0$, $\bar{r}^{i,j}(p, p_j, a) = \bar{r}^{j,i}(p_j, p, a) = 0$, $S^{i,j}(p, p_j, a) = S^{j,i}(p_j, p, a) = 0$ for all $a \in \mathcal{A}$, $j \in \mathcal{D}_{-i}$ and $p_j \in \mathcal{P}_{j,t}$.
3: **end for**

Figure 2: Pseudocode for ORL-CF.

Since the number of contexts is infinite, learning the reward of an action for each context is not feasible. In order to learn fast, ORL-CF exploits the similarities between the contexts of the relevant type given in Assumption 1 to estimate the rewards of the actions. The key to success of our algorithm is that this estimation is good enough. ORL-CF adaptively forms the partition of the space for each type in $\mathcal{D}$, where the partition for the context space of type $i$ at time $t$ is denoted by $\mathcal{P}_{i,t}$. All the elements of $\mathcal{P}_{i,t}$ are disjoint intervals of $\mathcal{X}_i = [0, 1]$ whose lengths are elements of the set $\{1, 2^{-1}, 2^{-2}, \ldots\}$.[4] An interval with length $2^{-l}$, $l \geq 0$ is called a level $l$ interval, and for an interval $p$, $l(p)$ denotes its level, $s(p)$ denotes its length. By convention, intervals are of the form $(a, b]$, with the only exception being the interval containing 0, which is of the form $[0, b]$.[5] Let $p_{i,t} \in \mathcal{P}_{i,t}$ be the interval that $x_{i,t}$ belongs to, $\boldsymbol{p}_t := (p_{1,t}, \ldots, p_{D,t})$ and $\boldsymbol{\mathcal{P}}_t := (\mathcal{P}_{1,t}, \ldots, \mathcal{P}_{D,t})$.

The pseudocode of ORL-CF is given in Fig. 2. ORL-CF starts with $\mathcal{P}_{i,1} = \{\mathcal{X}_i\} = \{[0,1]\}$ for each $i \in \mathcal{D}$. As time goes on and more contexts arrive for each type $i$, it divides $\mathcal{X}_i$ into smaller and smaller intervals. The idea is to combine the past observations made in an interval to form sample mean reward estimates for each interval, and use it to approximate the expected rewards of actions for contexts lying in these intervals. The intervals are created in a way to balance the variation of the sample mean rewards due to the number of past observations that are used to calculate them and the variation of the expected rewards in each interval.

We also call $\mathcal{P}_{i,t}$ the *set of active intervals* for type $i$ at time $t$. Since the partition of each type is adaptive, as time goes on, new intervals become active while old intervals are deactivated, based on how contexts arrive. For a type $i$ interval $p$, let $N_t^i(p)$ be the number of times $x_{i,t'} \in p \in \mathcal{P}_{i,t'}$ for $t' \leq t$. The duration of time that an interval remains active, i.e., its *lifetime*, is determined by an input parameter $\rho > 0$, which is called the *duration parameter*. Whenever the number of arrivals to an interval $p$ exceeds $2^{\rho l(p)}$, ORL-CF deactivates $p$ and creates two level $l(p)+1$ intervals, whose union gives $p$. For example, when $p_{i,t} = (k2^{-l}, (k+1)2^{-l}]$ for some $0 < k \leq 2^l - 1$ if $N_t^i(p_{i,t}) \geq 2^{\rho l}$, ORL-CF sets

$$\mathcal{P}_{i,t+1} = \mathcal{P}_{i,t} \cup \{(k2^{-l}, (k+1/2)2^{-l}], ((k+1/2)2^{-l}, (k+1)2^{-l}]\} - \{p_{i,t}\}.$$

Otherwise $\mathcal{P}_{i,t+1}$ remains the same as $\mathcal{P}_{i,t}$. It is easy to see that the lifetime of an interval increases exponentially in its duration parameter.

We next describe the counters, control numbers and sample mean rewards the learner keeps for each pair of intervals corresponding to a pair of types to determine whether to explore or exploit and how to exploit. Let $\mathcal{D}_{-i} := \mathcal{D} - \{i\}$. For type $i$, let $Q_{i,t} := \{(p_{i,t}, p_{j,t}) : j \in \mathcal{D}_{-i}\}$ be the pair of intervals that are *related to* type $i$ at time $t$, and let

$$Q_t := \bigcup_{i \in \mathcal{D}} Q_{i,t}. \tag{2}$$

To denote an element of $Q_{i,t}$ or $Q_t$ we use index $q$. For any $q \in Q_t$, the corresponding pair of types is denoted by $\mathrm{ind}(q)$. For example, $\mathrm{ind}((p_{i,t}, p_{j,t})) = i, j$. The decision to explore or exploit at time $t$ is solely based on $\boldsymbol{p}_t$. For events $A_1, \ldots, A_K$, let $\mathrm{I}(A_1, \ldots, A_k)$ denote the indicator function of event $\bigcap_{k=1:K} A_k$. For $p \in \mathcal{P}_{i,t}, p' \in \mathcal{P}_{j,t}$, let

$$S_t^{i,j}(p, p', a) := \sum_{t'=1}^{t-1} \mathrm{I}\left(\alpha_{t'} = a, \beta_t = 1, p_{i,t'} = p, p_{j,t'} = p'\right),$$

be the number of times $a$ is selected and the reward is observed when the type $i$ context is in $p$ and type $j$ context is in $p'$, summed over times when both intervals are active. Also for the same $p$ and $p'$ let

$$\bar{r}_t^{i,j}(p, p', a) := \left(\sum_{t'=1}^{t-1} r_t(a, \boldsymbol{x}_t) \mathrm{I}\left(\alpha_{t'} = a, \beta_t = 1, p_{i,t'} = p, p_{j,t'} = p'\right)\right) / (S_t^{i,j}(p, p', a)),$$

be the pairwise sample mean reward of action $a$ for pair of intervals $(p, p')$.

At time $t$, ORL-CF assigns a *control number* to each $i \in \mathcal{D}$ denoted by

$$D_{i,t} := \frac{2 \log(tD|\mathcal{A}|/\delta)}{(Ls(p_{i,t}))^2},$$

which depends on the cardinality of $\mathcal{A}$, the length of the active interval that type $i$ context is in at time $t$ and a *confidence parameter* $\delta > 0$, which controls the accuracy of sample mean reward estimates. Then, it computes the set of under-explored actions for type $i$ as

$$\mathcal{U}_{i,t} := \{a \in \mathcal{A} : S_t^{\mathrm{ind}(q)}(q, a) < D_{i,t} \text{ for some } q \in Q_i(t)\}, \tag{3}$$

and then, the set of under-explored actions as $\mathcal{U}_t := \bigcup_{i \in \mathcal{D}} \mathcal{U}_{i,t}$. The decision to explore or exploit is based on whether or not $\mathcal{U}_t$ is empty.

(i) If $\mathcal{U}_t \neq \emptyset$, ORL-CF randomly selects an action $\alpha_t \in \mathcal{U}_t$ to explore, and observes its reward $r_t(\alpha_t, \boldsymbol{x}_t)$. Then, it updates the pairwise sample mean rewards and pairwise counters for all $q \in Q_t$,

$$\bar{r}_{t+1}^{\mathrm{ind}(q)}(q, \alpha_t) = \frac{S_t^{\mathrm{ind}(q)}(q, \alpha_t) \bar{r}_{t+1}^{\mathrm{ind}(q)}(q, \alpha_t) + r_t(\alpha_t, \boldsymbol{x}_t)}{S_t^{\mathrm{ind}(q)}(q, \alpha_t) + 1}, \quad S_{t+1}^{\mathrm{ind}(q)}(q, \alpha_t) = S_t^{\mathrm{ind}(q)}(q, \alpha_t) + 1.$$

(ii) If $\mathcal{U}_t = \emptyset$, ORL-CF exploits by estimating the relevant type $\hat{c}_t(a)$ for each $a \in \mathcal{A}$ and forming sample mean reward estimates for action $a$ based on $\hat{c}_t(a)$. It first computes the set of *candidate relevant types* for each $a \in \mathcal{A}$,

$$\text{Rel}_t(a) := \{i \in \mathcal{D} : |\bar{r}_t^{i,j}(p_{i,t}, p_{j,t}, a) - \bar{r}_t^{i,k}(p_{i,t}, p_{k,t}, a)| \leq 3Ls(p_{i,t}), \forall j, k \in \mathcal{D}_{-i}\}. \quad (4)$$

The intuition is that if $i$ is the type that is relevant to $a$, then independent of the values of the contexts of the other types, the variation of the pairwise sample mean reward of $a$ over $p_{i,t}$ must be very close to the variation of the expected reward of $a$ in that interval.

If $\text{Rel}_t(a)$ is empty, this implies that ORL-CF failed to identify the relevant type, hence $\hat{c}_t(a)$ is randomly selected from $\mathcal{D}$. If $\text{Rel}_t(a)$ is nonempty, ORL-CF computes the maximum variation

$$\text{Var}_t(i, a) := \max_{j,k \in \mathcal{D}_{-i}} |\bar{r}_t^{i,j}(p_{i,t}, p_{j,t}, a) - \bar{r}_t^{i,k}(p_{i,t}, p_{k,t}, a)|, \quad (5)$$

for each $i \in \text{Rel}_t(a)$. Then it sets $\hat{c}_t(a) = \min_{i \in \text{Rel}_t(a)} \text{Var}_t(i, a)$. This way, whenever the type relevant to action $a$ is in $\text{Rel}_t(a)$, even if it is not selected as the estimated relevant type, the sample mean reward of $a$ calculated based on the estimated relevant type will be very close to the sample mean of its reward calculated according to the *true* relevant type. After finding the estimated relevant types, the sample mean reward of each action is computed based on its estimated relevant type as

$$\bar{r}_t^{\hat{c}_t(a)}(a) := \frac{\sum_{j \in \mathcal{D}_{-\hat{c}_t(a)}} \bar{r}_t^{\hat{c}_t(a),j}(p_{\hat{c}_t(a),t}, p_{j,t}, a) S_t^{\hat{c}_t(a),j}(p_{\hat{c}_t(a),t}, p_{j,t}, a)}{\sum_{j \in \mathcal{D}_{-\hat{c}_t(a)}} S_t^{\hat{c}_t(a),j}(p_{\hat{c}_t(a),t}, p_{j,t}, a)}. \quad (6)$$

Then, ORL-CF selects $\alpha_t = \arg\max_{a \in \mathcal{A}} \bar{r}_t^{\hat{c}_t(a)}(p_{\hat{c}_t(a),t}, a)$. Since the reward is not observed in exploitations, pairwise sample mean rewards and counters are not updated.

## 3.2 Regret analysis of ORL-CF

Let $\tau(T) \subset \{1, 2, \ldots, T\}$ be the set of time steps in which ORL-CF exploits by time $T$. $\tau(T)$ is a random set which depends on context arrivals and the randomness of the action selection of ORL-CF. The regret $R(T)$ defined in (1) can be written as a sum of the regret incurred during explorations (denoted by $R_O(T)$) and the regret incurred during exploitations (denoted by $R_I(T)$). The following theorem gives a bound on the regret of ORL-CF in exploitation steps.

**Theorem 1.** *Let ORL-CF run with duration parameter $\rho > 0$, confidence parameter $\delta > 0$ and control numbers $D_{i,t} := \frac{2\log(t|\mathcal{A}|D/\delta)}{(Ls(p_{i,t}))^2}$, for $i \in \mathcal{D}$. Let $R_{inst}(t)$ be the instantaneous regret at time $t$, which is the loss in expected reward at time $t$ due to not selecting $a^*(\boldsymbol{x}_t)$. Then, with probability at least $1 - \delta$, we have*

$$R_{inst}(t) \leq 8L(s(p_{\mathcal{R}(\alpha_t),t}) + s(p_{\mathcal{R}(a^*(\boldsymbol{x}_t)),t})),$$

*for all $t \in \tau(T)$, and the total regret in exploitation steps is bounded above by*

$$R_I(T) \leq 8L \sum_{t \in \tau(T)} (s(p_{\mathcal{R}(\alpha_t),t} + s(p_{\mathcal{R}(a^*(\boldsymbol{x}_t)),t})) \leq 16L2^{2\rho}T^{\rho/(1+\rho)},$$

*for arbitrary context vectors $\boldsymbol{x}_1, \boldsymbol{x}_2, \ldots, \boldsymbol{x}_T$.*

Theorem 1 provides both context arrival process dependent and worst case bounds on the exploitation regret of ORL-CF. By choosing $\rho$ arbitrarily close to zero, $R_I(T)$ can be made $O(T^\gamma)$ for any $\gamma > 0$. While this is true, the reduction in regret for smaller $\rho$ not only comes from increased accuracy, but it is also due to the reduction in the number of time steps in which ORL-CF exploits, i.e., $|\tau(T)|$. By definition time $t$ is an exploitation step if

$$S_t^{i,j}(p_{i,t}, p_{j,t}, a) \geq \frac{2\log(t|\mathcal{A}|D/\delta)}{L^2 \min\{s(p_{i,t})^2, s(p_{j,t})^2\}} = \frac{2^{2\max\{l(p_{i,t}), l(p_{j,t})\}+1}\log(t|\mathcal{A}|D/\delta)}{L^2},$$

for all $q = (p_{i,t}, p_{j,t}) \in Q_t$, $i, j \in \mathcal{D}$. This implies that for any $q \in Q_{i,t}$ which has the interval with maximum level equal to $l$, $\tilde{O}(2^{2l})$ explorations are required before any exploitation can take place. Since the time a level $l$ interval can stay active is $2^{\rho l}$, it is required that $\rho \geq 2$ so that $\tau(T)$ is nonempty.

The next theorem gives a bound on the regret of ORL-CF in exploration steps.

**Theorem 2.** *Let ORL-CF run with $\rho$, $\delta$ and $D_{i,t}$, $i \in \mathcal{D}$ values as stated in Theorem 1. Then,*

$$R_O(T) \leq \frac{960D^2(c_O + 1)\log(T|\mathcal{A}|D/\delta)}{7L^2}T^{4/\rho} + \frac{64D^2(c_O + 1)}{3}T^{2/\rho},$$

*with probability 1, for arbitrary context vectors $\boldsymbol{x}_1, \boldsymbol{x}_2, \ldots, \boldsymbol{x}_T$.*

Based on the choice of the duration parameter $\rho$, which determines how long an interval will stay active, it is possible to get different regret bounds for explorations and exploitations. Any $\rho > 4$ will give a sublinear regret bound for both explorations and exploitations. The regret in exploitations increases in $\rho$ while the regret in explorations decreases in $\rho$.

**Theorem 3.** *Let ORL-CF run with $\delta$ and $D_{i,t}$, $i \in \mathcal{D}$ values as stated in Theorem 1 and $\rho = 2 + 2\sqrt{2}$. Then, the time order of exploration and exploitation regrets are balanced up to logaritmic orders. With probability at least $1 - \delta$ we have both $R_I(T) = \tilde{O}(T^{2/(1+\sqrt{2})})$ and $R_O(T) = \tilde{O}(T^{2/(1+\sqrt{2})})$.*

**Remark 1.** *Prior work on contextual bandits focused on balancing the regret due to exploration and exploitation. For example in [1, 2], for a D-dimensional context vector algorithms are shown to achieve $\tilde{O}(T^{(D+1)/(D+2)})$ regret.[6] Also in [1] a $O(T^{(D+1)/(D+2)})$ lower bound on the regret is proved. An interesting question is to find the tightest lower bound for contextual bandits with relevance function. One trivial lower bound is $O(T^{2/3})$, which corresponds to $D = 1$. However, since finding the action with the highest expected reward for a context vector requires comparisons of estimated rewards of actions with different relevant types, which requires accurate sample mean reward estimates for 2 dimensions of the context space corresponding to those types, we conjecture that a tighter lower bound is $O(T^{3/4})$. Proving this is left as future work.*

Another interesting case is when actions with suboptimality greater than $\epsilon > 0$ must never be chosen in any exploitation step by time $T$. When such a condition is imposed, ORL-CF can start with partitions $\mathcal{P}_{i,1}$ that have sets with high levels such that it explores more at the beginning to have more accurate reward estimates before any exploitation. The following theorem gives the regret bound of ORL-CF for this case.

**Theorem 4.** *Let ORL-CF run with duration parameter $\rho > 0$, confidence parameter $\delta > 0$, control numbers $D_{i,t} := \frac{2\log(t|\mathcal{A}|D/\delta)}{(Ls(p_{i,t}))^2}$, and with initial partitions $\mathcal{P}_{i,1}$, $i \in \mathcal{D}$ consisting of intervals of length $l_{\min} = \lceil \log_2(3L/(2\epsilon)) \rceil$. Then, with probability $1 - \delta$, $R_{inst}(t) \leq \epsilon$ for all $t \in \tau(T)$, $R_I(T) \leq 16L2^{2\rho}T^{\rho/(1+\rho)}$ and*

$$R_O(T) \leq \frac{81L^4}{\epsilon^4}\left(\frac{960D^2(c_O + 1)\log(T|\mathcal{A}|D/\delta)}{7L^2}T^{4/\rho} + \frac{64D^2(c_O + 1)}{3}T^{2/\rho}\right),$$

*for arbitrary context vectors $\boldsymbol{x}_1, \boldsymbol{x}_2, \ldots, \boldsymbol{x}_T$. Bounds on $R_I(T)$ and $R_O(T)$ are balanced for $\rho = 2 + 2\sqrt{2}$.*

### 3.3 Future Work

In this paper we only considered the relevance relations that are functions. Similar learning methods can be developed for more general relevance relations such as the ones given in Fig. 1 $(i)$ and $(ii)$. For example, for the general case in Fig. 1 $(i)$, if $|\mathcal{R}(a)| \leq D_{\text{rel}} << D$, for all $a \in \mathcal{A}$, and $D_{\text{rel}}$ is known by the learner, the following variant of ORL-CF can be used to achieve regret whose time order depends only on $D_{\text{rel}}$ but not on $D$.

- Instead of keeping pairwise sample mean reward estimates, keep sample mean reward estimates of actions for $D_{\text{rel}} + 1$ tuples of intervals of $D_{\text{rel}} + 1$ types.

- For a $D_{\text{rel}}$ tuple of types $\boldsymbol{i}$, let $Q_{\boldsymbol{i},t}$ be the $D_{\text{rel}} + 1$ tuples of intervals that are related to $\boldsymbol{i}$ at time $t$, and $Q_t$ be the union of $Q_{\boldsymbol{i},t}$ over all $D_{\text{rel}}$ tuples of types. Similar to ORL-CF, compute the set of under-explored actions $\mathcal{U}_{\boldsymbol{i},t}$, and the set of candidate relevant $D_{\text{rel}}$ tuples of types $\text{Rel}_t(a)$, using the newly defined sample mean reward estimates.

- In exploitation, set $\hat{c}_t(a)$ to be the $D_{\text{rel}}$ tuple of types with the minimax variation, where the variation of action a for a tuple $\boldsymbol{i}$ is defined similar to (5), as the maximum of the distance between the sample mean rewards of action $a$ for $D_{\text{rel}}+1$ tuples that are in $Q_{\boldsymbol{i},t}$.

Another interesting case is when the relevance relation is linear as given in Fig. 1 $(ii)$. For example, for action $a$ if there is a type $i$ that is much more relevant compared to other types $j \in \mathcal{D}_{-i}$, i.e., $w_{a,i} >> w_{a,j}$, where the weights $w_{a,i}$ are given in Fig. 1, then ORL-CF is expected to have good performance (but not sublinear regret with respect to the benchmark that knows $\mathcal{R}$).

## 4 Related Work

Contextual bandit problems are studied by many others in the past [3, 4, 1, 2, 5, 6]. The problem we consider in this paper is a special case of the Lipschitz contextual bandit problem [1, 2], where the only assumption is the existence of a known similarity metric between the expected rewards of actions for different contexts. It is known that the lower bound on regret for this problem is $O(T^{(D+1)/(D+2)})$ [1], and there exists algorithms that achieve $\tilde{O}(T^{(D+1)/(D+2)})$ regret [1, 2]. Compared to the prior work above, ORL-CF only needs to observe rewards in explorations and has a regret whose time order is independent of $D$. Hence it can still learn the optimal actions fast enough in settings where observations are costly and context vector is high dimensional.

Examples of related works that consider limited observations are KWIK learning [7, 8] and label efficient learning [9, 10, 11]. For example, [8] considers a bandit model where the reward function comes from a parameterized family of functions and gives bound on the average regret. An online prediction problem is considered in [9, 10, 11], where the predictor (action) lies in a class of linear predictors. The benchmark of the context is the best linear predictor. This restriction plays a crucial role in deriving regret bounds whose time order does not depend on $D$. Similar to these works, ORL-CF can guarantee with a high probability that actions with large suboptimalities will never be selected in exploitation steps. However, we do not have any assumptions on the form of the expected reward function other than the Lipschitz continuity and that it depends on a single type for each action.

In [12] graphical bandits are proposed where the learner takes an action vector $\boldsymbol{a}$ which includes actions from several types that consitute a type set $\mathcal{T}$. The expected reward of $\boldsymbol{a}$ for context vector $\boldsymbol{x}$ can be decomposed into sum of reward functions each of which only depends on a subset of $\mathcal{D} \cup \mathcal{T}$. However, it is assumed that the form of decomposition is known but the functions are not known. Another work [13] proposes a fast learning algorithm for an i.i.d. contextual bandit problem in which the rewards for contexts and actions are sampled from a joint probability distribution. In this work the authors consider learning the best policy from a finite set of policies with oracle access, and prove a regret bound of $O(\sqrt{T})$ which is also logarithmic in the size of the policy space. In contrast, in our problem $(i)$ contexts arrive according to an arbitrary exogenous process, and the action rewards are sampled from an i.i.d. distribution given the context value, $(ii)$ the set of policies that the learner can adopt is not restricted.

Large dimensional action spaces, where the rewards depend on a subset of the types of actions are considered in [14] and [15]. [14] considers the problem when the reward is Hölder continuous in an unknown low-dimensional tuple of types, and uses a special discretization of the action space to achieve dimension independent bounds on the regret. This discretization can be effectively used since the learner can select the actions, as opposed to our case where the learner does not have any control over contexts. [15] considers the problem of optimizing high dimensional functions that have an unknown low dimensional structure from noisy observations.

## 5 Conclusion

In this paper we formalized the problem of learning the best action through learning the relevance relation between types of contexts and actions. For the case when the relevance relation is a function, we proposed an algorithm that $(i)$ has sublinear regret with time order independent of $D$, $(ii)$ only requires reward observations in explorations, $(iii)$ for any $\epsilon > 0$, does not select any $\epsilon$ suboptimal actions in exploitations with a high probability. In the future we will extend our results to the linear and general relevance relations illustrated in Fig. 1.

## Footnotes

[1] For a set $\mathcal{A}$, $|\mathcal{A}|$ denotes its cardinality.

[2]Even when there are multiple relevant types for each action, but there is one dominant type whose effect on the reward of the action is significantly larger than the effects of other types, assuming that the relevance relation is a function will be a good approximation.

[3]$O(\cdot)$ is the Big O notation, $\tilde{O}(\cdot)$ is the same as $O(\cdot)$ except it hides terms that have polylogarithmic growth.

[4]Setting interval lengths to powers of 2 is for presentational simplicity. In general, interval lengths can be set to powers of any real number greater than 1.

[5]Endpoints of intervals will not matter in our analysis, so our results will hold even when the intervals have common endpoints.

[6]The results are shown in terms of the *covering dimension* which reduces to Euclidian dimension for our problem.

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
