[Supplementary Material · supplementalfinal.pdf]

# SUPPLEMENTAL MATERIAL
# of
# Discovering, Learning and Exploiting Relevance

**Cem Tekin**
Electrical Engineering Department
University of California Los Angeles
cmtkn@ucla.edu

**Mihaela van der Schaar**
Electrical Engineering Department
University of California Los Angeles
mihaela@ee.ucla.edu

This supplementary material is composed of two sections. The first section includes numerical results on the proposed algorithm. The second section includes proofs of the theorems.

## 1 Experiments

### 1.1 The dataset

The network intrusion dataset from UCI archive [1] consists of a series of TCP connection records, labeled either as normal connections or as attacks. The data consists of 42 features, and we take 15 of them as types of contexts. Taken features are normalized to lie in $[0, 1]$. The prediction action belongs to the set $\{attack, noattack\}$. Reward is 1 when the prediction is correct and 0 otherwise.

### 1.2 Learning methods that we compare against

**Contextual zooming (CZ)** [2]: This algorithm adaptively creates balls over the joint action and context space, calculates an index for each ball based on the history of selections of that ball, and at each time step selects an action according to the ball with the highest index that contains the action-context pair.

**Hybrid-$\epsilon$** [3]: This algorithm is the contextual version of $\epsilon$-greedy, which forms context-dependent sample mean rewards for the actions by considering the history of observations and decisions for groups of contexts that are similar to each other.

**Ensemble Learning Methods** Average Majority (AM) [5], Adaboost [6], Online Adaboost [7] and Blum's Variant of Weighted Majority (Blum) [8]: The goal of ensemble learning is to create a strong (high accuracy) classifier by combining predictions of base classifiers. Hence all these methods require base classifiers (trained a priori) that produce predictions (or actions) based on the context vector.

AM simply follows the prediction of the majority of the classifiers and does not perform active learning. Adaboost is trained a priori with 1500 instances, whose labels are used to compute the weight vector. Its weight vector is fixed during the test phase (it is not learning online); hence no active learning is performed during the test phase. In contrast, Online Adaboost always receives the true label at the end of each time slot. It uses a time window of 1000 past observations to retrain its weight vector. Similar to Online Adaboost, Blum also learns its weight vector online.

### 1.3 Numerical Results

We compare the performance of ORL-CF with other learning methods described in the previous subsection. For the ensemble learning methods, the base classifiers are logistic regression classifiers, each trained with 5000 different instances from the dataset. Comparison of performances in terms of the error rate is given in Table 1. We see that ORL-CF has the lowest error rate at $1.19\%$, which is $27\%$ less than the second best method (Blum). All the ensemble learning methods we compare

against use classifiers to make predictions, and these classifiers require a priori training. In contrast, ORL-CF do not require any a priori training and learns online.

| Algorithm | Reference | error % |
|---|---|---|
| AM | [5] | 3.07 |
| Adaboost | [6] | 3.1 |
| Online Adaboost | [7] | 2.25 |
| Blum | [8] | 1.64 |
| CZ | [2] | 53 |
| Hybrid-$\epsilon$ | [3] | 8.8 |
| ORL-CF | our work | 1.19 |

Table 1: Comparison of the error rate of ORL-CF with ensemble learning methods and other online learning methods for the network intrusion dataset.

## 2 Proofs

Let $A := |\mathcal{A}|$. We first define a sequence of events which will be used in the analysis of the regret of ORL-CF. For $p \in \mathcal{P}_{\mathcal{R}(a),t}$, Let $\pi(a,p) = \mu(a, x^*_{\mathcal{R}(a)}(p))$, where $x^*_{\mathcal{R}(a)}(p)$ is the context at the geometric center of $p$. For $j \in \mathcal{D}_{-\mathcal{R}(a)}$, let

$$\text{INACC}_t(a,j) := \left\{ |\bar{r}_t^{\mathcal{R}(a),j}(p_{\mathcal{R}(a),t}, p_{j,t}, a) - \pi(a, p_{\mathcal{R}(a),t})| > \frac{3}{2} Ls(p_{\mathcal{R}(a),t}) \right\},$$

be the event that the pairwise sample mean corresponding to pair $(\mathcal{R}(a), j)$ of types is *inaccurate* for action $a$. Let

$$\text{ACC}_t(a) := \bigcap_{j \in \mathcal{D}_{-\mathcal{R}(a)}} \text{INACC}_t(a,j)^C,$$

be the event that all pairwise sample means corresponding to pairs $(\mathcal{R}(a), j)$, $j \in \mathcal{D}_{-\mathcal{R}(a)}$ are accurate. Consider $t \in \tau(T)$. Let

$$\text{WNG}_t(a) := \{ \mathcal{R}(a) \notin \text{Rel}_t(a) \},$$

be the event that the type relevant to action $a$ is not in the set of candidate relevant types, and

$$\text{WNG}_t := \bigcup_{a \in \mathcal{A}} \text{WNG}_t(a),$$

be the event that the type relevant to some action $a$ is not in the set of candidate relevant types of that action. Finally, let

$$\text{CORR}_T := \bigcap_{t \in \tau(T)} \text{WNG}_t^C,$$

be the event that the relevant types for all actions are in the set of candidate relevant types at all exploitation steps.

### 2.1 Proof of Theorem 1

We first prove several lemmas related to Theorem 1. The next lemma gives a lower bound on the probability of $\text{CORR}_T$.

**Lemma 1.** *For ORL-CF, for all $a \in \mathcal{A}$, $t \in \tau(T)$, we have*

$$\text{P}(\textit{INACC}_t(a,j)) \leq \frac{2\delta}{ADt^4}.$$

*for all $j \in \mathcal{D}_{-\mathcal{R}(a)}$, and $\text{P}(\textit{CORR}_T) \geq 1 - \delta$ for any $T$.*

*Proof.* For $t \in \tau(T)$, we have $\mathcal{U}_t = \emptyset$, hence

$$S_t^{\text{ind}(q)}(q,a) \geq \frac{2 \log(tAD/\delta)}{(Ls(p_{\mathcal{R}(a),t}))^2},$$

for all $a \in \mathcal{A}$, $q \in Q_i(t)$ and $i \in \mathcal{D}$. Due to Assumption 1, since rewards in $\bar{r}_t^{\mathcal{R}(a),j}(p_{\mathcal{R}(a),t}, p_{j,t}, a)$ are sampled from distributions with mean between $[\pi(a, p_{\mathcal{R}(a),t}) - \frac{L}{2}s(p_{\mathcal{R}(a),t}), \pi(a, p_{\mathcal{R}(a),t}) + \frac{L}{2}s(p_{\mathcal{R}(a),t})]$, using a Chernoff bound we get

$$\begin{aligned}
\mathrm{P}(\mathrm{INACC}_t(a,j)) &\leq 2\exp\left(-2(Ls(p_{\mathcal{R}(a),t}))^2 \frac{2\log(tAD/\delta)}{(Ls(p_{\mathcal{R}(a),t}))^2}\right) \\
&= 2\exp\left(-4(\log t + \log(AD/\delta))\right) \\
&\leq 2\exp\left(-4(\log t)\right)\exp\left(-\log(AD/\delta)\right) \\
&\leq \frac{2\delta}{ADt^4}.
\end{aligned}$$

We have $\mathrm{WNG}_t(a) \subset \bigcup_{j \in \mathcal{D}_{-\mathcal{R}(a)}} \mathrm{INACC}_t(a,j)$. Thus

$$\mathrm{P}(\mathrm{WNG}_t(a)) \leq \frac{2\delta}{At^4}, \text{ and } \mathrm{P}(\mathrm{WNG}_t) \leq \frac{2\delta}{t^4}.$$

This implies that

$$\mathrm{P}(\mathrm{CORR}_T^C) \leq \sum_{t \in \tau(T)} \mathrm{P}(\mathrm{WNG}_t) \tag{1}$$

$$\leq \sum_{t \in \tau(T)} \frac{2\delta}{t^4} \leq \sum_{t=3}^{\infty} \frac{2\delta}{t^4} \leq \delta.$$

$\square$

**Lemma 2.** *When CORR$_T$ happens we have for all $t \in \tau(T)$*

$$|\bar{r}_t^{\hat{c}_t(a)}(p_{\hat{c}_t(a),t}, a) - \mu(a, x_{\mathcal{R}(a),t})| \leq 8Ls(p_{\mathcal{R}(a),t}).$$

*Proof.* From Lemma 1, CORR$_T$ happens when

$$|\bar{r}_t^{\mathcal{R}(a),j}(p_{\mathcal{R}(a),t}, p_{j,t}, a) - \pi(a, p_{\mathcal{R}(a),t})| \leq \frac{3L}{2}s(p_{\mathcal{R}(a),t}),$$

for all $a \in \mathcal{A}$, $j \in \mathcal{D}_{-\mathcal{R}(a)}$, $t \in \tau(T)$. Since $|\mu(a, x_{\mathcal{R}(a),t}) - \pi(a, p_{\mathcal{R}(a),t})| \leq Ls(p_{\mathcal{R}(a),t})/2$, we have

$$|\bar{r}_t^{\mathcal{R}(a),j}(p_{\mathcal{R}(a),t}, p_{j,t}, a) - \mu(a, x_{\mathcal{R}(a),t})| \leq 2Ls(p_{\mathcal{R}(a),t}), \tag{2}$$

for all $a \in \mathcal{A}$, $j \in \mathcal{D}_{-\mathcal{R}(a)}$, $t \in \tau(T)$. Consider $\hat{c}_t(a)$. Since it is chosen from $\mathrm{Rel}_t(a)$ as the type with the minimum variation, we have on the event CORR$_T$

$$|\bar{r}_t^{\hat{c}_t(a),k}(p_{\hat{c}_t(a),t}, p_{k,t}, a) - \bar{r}_t^{\hat{c}_t(a),j}(p_{\hat{c}_t(a),t}, p_{j,t}, a)| \leq 3Ls(p_{\mathcal{R}(a),t}),$$

for all $j, k \in \mathcal{D}_{-\hat{c}_t(a)}$. Hence we have

$$\begin{aligned}
&|\bar{r}_t^{\mathcal{R}(a)}(p_{\mathcal{R}(a),t}, a) - \bar{r}_t^{\hat{c}_t(a)}(p_{\hat{c}_t(a),t}, a)| \\
&\leq \max_{k,j} |\bar{r}_t^{\mathcal{R}(a),k}(p_{\mathcal{R}(a),t}, p_{k,t}, a) - \bar{r}_t^{\hat{c}_t(a),j}(p_{\hat{c}_t(a),t}, p_{j,t}, a)| \\
&\leq \max_{k,j} \left(|\bar{r}_t^{\mathcal{R}(a),k}(p_{\mathcal{R}(a),t}, p_{k,t}, a) - \bar{r}_t^{\mathcal{R}(a),\hat{c}_t(a)}(p_{\mathcal{R}(a),t}, p_{\hat{c}_t(a),t}, a)| \right.\\
&\left.+|\bar{r}_t^{\hat{c}_t(a),\mathcal{R}(a)}(p_{\hat{c}_t(a),t}, p_{\mathcal{R}(a),t}, a) - \bar{r}_t^{\hat{c}_t(a),j}(p_{\hat{c}_t(a),t}, p_{j,t}, a)|\right) \\
&\leq 6Ls(p_{\mathcal{R}(a),t}). \tag{3}
\end{aligned}$$

Combining 2 and 3, we get

$$|\bar{r}_t^{\hat{c}_t(a)}(p_{\hat{c}_t(a),t}, a) - \mu(a, x_{\mathcal{R}(a),t})| \leq 8Ls(p_{\mathcal{R}(a),t}).$$

$\square$

Since for $t \in \tau(T)$, $\alpha_t = \arg\max_{a \in \mathcal{A}} \bar{r}_t^{\hat{c}_t(a)}(p_{\hat{c}_t(a),t}, a)$, using the result of Lemma 2, we conclude that

$$\mu_t(\alpha_t) \geq \mu_t(a^*(\boldsymbol{x}_t)) - 8L(s(p_{\mathcal{R}(\alpha_t),t}) + s(p_{\mathcal{R}(a^*(\boldsymbol{x}_t)),t})), \tag{4}$$

Thus, the regret in exploitation steps is

$$8L \sum_{t \in \tau(T)} \left( s(p_{\mathcal{R}(\alpha_t),t}) + s(p_{\mathcal{R}(a^*(\boldsymbol{x}_t)),t}) \right) \leq 16L \sum_{t \in \tau(T)} \max_{a \in \mathcal{A}} s(p_{\mathcal{R}(a),t})$$

$$\leq 16L \sum_{t \in \tau(T)} \max_{i \in \mathcal{D}} s(p_{i,t})$$

$$\leq 16L \sum_{t \in \tau(T)} \sum_{i \in \mathcal{D}} s(p_{i,t})$$

$$\leq 16L \sum_{i \in \mathcal{D}} \max_{i \in \mathcal{D}} \left( \sum_{t \in \tau(T)} s(p_{i,t}) \right)$$

$$= 16LD \max_{i \in \mathcal{D}} \left( \sum_{t \in \tau(T)} s(p_{i,t}) \right).$$

We know that as time goes on ORL-CF uses partitions with smaller and smaller intervals, which reduces the regret in exploitations. In order to bound the regret in exploitations for any sequence of context arrivals, we assume a worst case scenario, where context vectors arrive such that at each $t$, the active interval that contains the context of each type has the maximum possible length. This happens when for each type $i$ contexts arrive in a way that all level $l$ intervals are split to level $l+1$ intervals, before any arrivals to these level $l+1$ intervals happen, for all $l = 0, 1, 2, \ldots$. This way it is guaranteed that the length of the interval that contains the context for each $t \in \tau(T)$ is maximized. Let $l_{\max}$ be the level of the maximum level interval in $\mathcal{P}_i(T)$. For the worst case context arrivals we must have

$$\sum_{l=0}^{l_{\max}-1} 2^l 2^{\rho l} < T$$

$$\Rightarrow l_{\max} < 1 + \log_2 T/(1+\rho),$$

since otherwise maximum level hypercube will have level larger than $l_{\max}$. Hence we have

$$16LD \max_{i \in \mathcal{D}} \left( \sum_{t \in \tau(T)} s(p_{i,t}) \right) \leq 16L \sum_{l=0}^{1+\log_2 T/(1+\rho)} 2^l 2^{\rho l} 2^{-l}$$

$$= 16L \sum_{l=0}^{1+\log_2 T/(1+\rho)} 2^{\rho l}$$

$$\leq 16L 2^{2\rho} T^{\rho/(1+\rho)}. \tag{5}$$

## 2.2 Proof of Theorem 2

Recall that time $t$ is an exploitation step only if $\mathcal{U}_t = \emptyset$. In order for this to happen we need $S_t^{i,j}(p_{i,t}, p_{j,t}, a) \geq D_{i,t}$ for all $q \in Q_i(t)$. Since for any $p_i \in \mathcal{P}_{i,t}$, $p_j \in \mathcal{P}_{j,t}$ we have $S_t^{i,j}(p_i, p_j, a) = S_t^{j,i}(p_j, p_i, a)$, the number of explorations of pair $(p_i, p_j)$ at time $t$ will be at most

$$\frac{2\log(tAD/\delta)}{L^2 \min(s(p_i), s(p_j))^2} + 1 \tag{6}$$

There are $D(D-1)$ type pairs. Whenever action $a$ is explored, all the counters for these $D(D-1)$ type pairs are updated for the pairs of intervals that contain types of contexts present at time $t$, i.e. $q \in Q_t$. Now consider a hypothetical scenario in which instead of updating the counters of all

$q \in Q_t$, the counter of only one of the randomly selected interval pair is updated. Clearly, the exploration regret of this hypothetical scenario upper bounds the exploration regret of the original scenario. We can go one step further and consider a second hypothetical scenario where there is only two types $i$ and $j$, for which the actual regret at every exploration step is magnified (multiplied) by $D(D-1)$. The maximum possible exploration regret of the second scenario (for the worst case of type $i$ and $j$ context arrivals) upper bounds the exploration regret of the first scenario. Hence, we bound the regret of the second scenario. Let $l_{\max}$ be the maximum possible level for an active interval for type $i$ by time $T$. We must have

$$\sum_{l=0}^{l_{\max}-1} 2^{\rho l} < T,$$

which implies that $l_{\max} < 1 + \log_2 T/\rho$. Next, we consider all pairs of intervals for which the minimum interval has level $l$. For each type $j$ interval $p_j$ that has level $l$, there exists no more than $\sum_{k=l}^{l_{\max}} 2^k$ type $i$ intervals that have lengths greater than or equal to $l$. Consider a level $k$ type $i$ interval $p_i$ such that $l \le k < 1 + \log_2 T/\rho$. Then for the pair of intervals $(p_i, p_j)$ the exploration regret is bounded by $(c_O + 1)\left(2\log(tAD/\delta)/(2^{-2k}L^2) + 1\right)$. Hence, the worst case exploration regret is bounded by

$$R_O(T) \le (c_O+1)D^2 \left( 2 \sum_{l=0}^{1+\log_2 T/\rho} 2^l \sum_{k=l}^{1+\log_2 T/\rho} 2^k \left( \frac{2\log(tAD/\delta)}{2^{-2k}L^2} + 1 \right) \right)$$

$$= (c_O+1)D^2 \left( \frac{4\log(tAD/\delta)}{L^2} \sum_{l=0}^{1+\log_2 T/\rho} 2^l \sum_{k=l}^{1+\log_2 T/\rho} 2^{3k} + 2 \sum_{l=0}^{1+\log_2 T/\rho} 2^l \sum_{k=l}^{1+\log_2 T/\rho} 2^k \right)$$

$$\le \frac{4D^2(c_O+1)\log(tAD/\delta)}{L^2} \times \frac{240}{7}T^{4/\rho} + \frac{64D^2(c_O+1)}{3}T^{2/\rho}.$$

### 2.3 Proof of Theorem 4

To achieve $\epsilon$-optimality in every exploitation step it is sufficient to have

$$\text{INACC}_t(a,j)^C = \left\{ |\bar{r}_t^{\mathcal{R}(a),j}(p_{\mathcal{R}(a),t}, p_{j,t}, a) - \pi(a, p_{\mathcal{R}(a),t})| < \frac{3}{2}Ls(p_{\mathcal{R}(a),t}) \right\},$$

$$\subset \left\{ |\bar{r}_t^{\mathcal{R}(a),j}(p_{\mathcal{R}(a),t}, p_{j,t}, a) - \pi(a, p_{\mathcal{R}(a),t})| < \epsilon \right\},$$

for $t \in \tau(T)$. This is satisfied when $l_{\min} \ge \log_2(3L/(2\epsilon))$. Starting with level $l_{\min}$ intervals instead of level 0 intervals decreases the exploitation regret of ORL-CF. Hence the regret bound in Theorem 1 is an upper bound on the exploitation regret.

For any sequence of context arrivals, we have the following bound on the level of the interval with the maximum level,

$$l_{\max} < 1 + l_{\min} + \log_2 T/\rho.$$

Continuing similarly with the proof of Theorem 2, we have

$$R_O(T) \le (c_O+1)D^2 \left( 2 \sum_{l=0}^{1+\log_2 T/\rho} 2^{l_{\min}} 2^l \sum_{k=l}^{1+\log_2 T/\rho} 2^{l_{\min}} 2^k \left( 2^{4l_{\min}} \frac{2\log(tAD/\delta)}{2^{-2l_{\min}}2^{-2k}L^2} + 1 \right) \right)$$

$$= (c_O+1)D^2 \left( \frac{4\log(tAD/\delta)}{L^2} \sum_{l=0}^{1+\log_2 T/\rho} 2^l \sum_{k=l}^{1+\log_2 T/\rho} 2^{3k} + 2^{2l_{\min}} 2 \sum_{l=0}^{1+\log_2 T/\rho} 2^l \sum_{k=l}^{1+\log_2 T/\rho} 2^k \right)$$

$$\le 2^{4l_{\min}} \left( \frac{4D^2(c_O+1)\log(tAD/\delta)}{L^2} \times \frac{240}{7}T^{4/\rho} + \frac{64D^2(c_O+1)}{3}T^{2/\rho} \right).$$