[Reviews · NeurIPS 2014]

Submitted by Assigned_Reviewer_16

This paper studies the multi-armed bandit problem where they have a set of relevant features; and the expected reward of an action is a Lipschitz continuous of relevant features. This is also a feature selection problem where you have a set of features but only r of them are relevant (the target function only depends on r of these features): here each arm has only one relevant feature, meaning the function representing the arm payoff depending on only one feature and we do not know which one. They propose an algorithm and get the bound for such adaptive case; but their regret is higher than what you would get if someone tells you the relevant type.

Summary: This paper makes a small step towards understanding the problem of having a subset of features being relevant for a given arm which itself is certainly an interesting problem: they study the bandit problem only for one relevant feature per arm and did not give the optimal rate. Potentially, they could go with all arbitrary number of relevant features and figure out the optimal regret.

Submitted by Assigned_Reviewer_40

The paper deals with the problem of contextual bandits in the following setting: The context is composed of the product of D continuous sets, which are throughout considered (w.l.o.g) to be the interval [0,1]. The gain of an arm is assumed to be a random variable with mean equal to some Lipschitz function that depends on exactly one of the D intervals. The contexts are assumed to be arbitrary. In each round the player chooses an arm and gains its defined gain. The player may choose to query the value of that gain at a cost. Eventually, the objective is to minimize the regret defined as the sum of gains of the optimal strategy minus the sum of gains earned by the player plus the sum of exploration costs.

The novelty w.r.t the setting is the sparsity of the function, meaning the fact that it depends on only one interval rather than all of them. This additional assumption is very natural and has been considered in other scenarios as well. Having it allows a significant increase in the performance of the algorithm. Specifically, it allows the authors to obtain a regret of T^c with constant c that is independent of the context dimension D (without the assumption there is a lower bound of c \geq (D+1)/(D+2)). In order to deal with the continuous regime, the authors use a (somewhat) standard technique of maintaining partitions of the interval with adaptively shrinking lengths. To deal with the sparsity of the gain function, the algorithm maintains, for each arm, estimates of the function for pairs of intervals originating from different intervals and determines the relevant interval (i.e., the interval that determines the value of the arm) by finding the one whose expected value is the same for all pairs containing it. Overall, the techniques are mostly technical but are definitely non-trivial.

The authors provide experiments in a setting where algorithm is used to combine several different classifiers. These experiments do not do justice with the main contribution of the paper since the dimension of the context is chosen to be 3. I would expect the experiments to compare the current algorithm with those that do not assume sparsity, and an advantage would only occur for large values of D. In particular, for D=3, previous results provide a regret of T^{0.8} while this paper guarantees a regret of T^{0.83}.

Comments:
• line 253: D should be |D|
• Line 361: The paper should state the exact dependence on D_{rel}
• The following paper could be relevant: Agarwal, Alekh, et al. "Taming the Monster: A Fast and Simple Algorithm for Contextual Bandits.". In particular, it would be interesting to see whether the mentioned setting can be solved using their techniques as well as the resulting regret.
• Appendix, line 103: log(tAD/\delta) should be log(t^4AD/\delta)
• Appendix, line 161, missing L

Summary: The paper deals with an interesting setting and provides an efficient solution for it. The methods used are mostly technical but non-trivial and interesting enough to justify acceptance.

Submitted by Assigned_Reviewer_42

Summary of the paper:
The paper considers a contextual bandit problem, with costly observations, and focuses on the case when reward distributions depend on a few parameters, called types, in way that is summarized by a so-called (unknown) relevance relation.
The paper introduces an algorithm (Section 3.1) to deal with this situation, that alternates between exploration and exploitation phases.
The regret corresponding to each phase is bounded in Theorem 1 and Theorem 2, and the total regret of the algorithm is shown to achieve order \tilde O(T^{2/(1+sqrt{2})}) in Theorem 3, for suitable parameter choice.

Main comments:
The paper looks interesting and sound. I only have minor comments.

In figure 1, I suggest you explicit mention that the focus on this paper is on (iii); otherwise it is misleading.

The introduction of (3) seems related in spirit to the definition of saturated arms in the analysis of Thompson sampling algorithm.
I am not sure it is needed to comment this, but more generally, it would be nice to provide some additional connection between the proof technique involved here and more standard ones. A sketch of proof may be valuable too.

The introduction can be made more precise. The notion of relevance is fuzzy until section 2.

L.127: "an" is duplicated.

Decision:
Accept.
----
I have read the feedback from the authors.
Summary: Possibly interesting paper, that requires some clarifications.
Author Feedback
Author rebuttal: Reviewer ID 16:

1) Thanks! Our main motivation for considering only a single relevant type of context (D_rel =1) for each arm was its presentation simplicity, which allowed us to highlight the key ideas in a limited number of pages. Moreover, although the general case (D_rel >=1) requires more involved notation, it does not require significantly different techniques and algorithms to establish the same strand of results. However, we do agree with the reviewer that many data sets exhibit general relevance relationships. Hence, we will modify the paper and change the learning algorithm (ORL-CF) to enable learning of general relevance relations, including relevance relations involving multiple types of relevant contexts (D_rel >= 1). While we already provided a sketch of this general algorithm in Section 3.3 of the original manuscript, based on the reviewer’s comment we will modify the paper to present this new, general algorithm and the associated theorem that bounds its regret. A key difference is the identification of multiple relevant types for each action, which requires estimating the variations of the expected arm rewards for different (D_rel+1) tuples of types to identify and rule out the set of types whose estimated rewards contradicts the Lipschitz similarity condition. An additional merit of the algorithm is that it also learns the relevance relation while maximizing the total reward.

Reviewer ID 40:

1) Thanks! We agree with the reviewer and in the final version of the paper we will include a new set of experiments in which we take a vector of D=15 features from a network security data set and a vector of D=9 features from a breast cancer data set and present our results. Both these data sets exhibit relevance relationships where the true label is highly correlated with a single type of context. Moreover, we will also simulate our algorithm’s performance for the scenario where no base classifiers are used (i.e. the algorithm will directly make predictions based solely on the values of the features). The experiments which we have performed so far show significant performance improvements in terms of prediction accuracy (e.g. over 40% decrease in the error rate for the breast cancer data set) as compared to the benchmark contextual bandit algorithms. As the reviewer pointed out, when the relevance relation is a function (D_rel = 1), our regret bound is better than the prior regret bound of O(T^{D+1/(D+2)}) for D>3. For instance when D=15, the previous results have a regret bound of O(T^{0.94}), while our algorithm will have a regret bound O(T^{0.828})

2) While [1] is a very interesting paper, it makes a different set of assumptions and provides very different techniques.
In [1]: Finite set of policies with oracle access. Our work: Infinitely many policies, similarity information.
In [1]: Contexts and actions are sampled from a joint probability distribution (iid contextual bandit problem). Our work: Contexts arrive according to an arbitrary exogenous process. Given the context, action rewards are iid.
In [1]: Regret bound is O(T^{1/2}) but also logarithmic in the size of the policy space. Our case: Regret bound is independent of the context space dimension because of the relevance relationship. Size of the policy space can grow exponentially with the number of contexts which can grow exponentially with the dimension of the context space.
In the final manuscript a discussion about the similarities and differences between [1] and our work will be included. Moreover, future work could combine the advantages of these two methods.

[1] Agarwal, Alekh, et al. “Taming the Monster: A Fast and Simple Algorithm For Contextual Bandits”

3) As we mentioned in the response to reviewer ID 16, we will modify the paper to present the algorithm and regret bounds for the general case when there are multiple relevant types for each action (D_rel >=1).

4) Thanks, all typos will be corrected.

Reviewer ID 42:

1) We will revise Figure 1 to clarify the goal of the paper. Moreover, we will also modify the paper and present the algorithm and associated regret bounds for the general case, when there are multiple relevant types for each action (D_rel >=1).

2) In Equation 3, U_{i,t} is the set of under-explored arms for the type i context at time t. These are the arms for which the learner’s confidence on its sample mean reward estimates is below the desired level. Hence, there is a high probability that the learner will fail to identify the relevant type correctly for these arms. They must be explored before the learner starts exploiting the arm with the highest sample mean reward for its estimated relevant context. The arms that are not in U_{i,t} at time t are sufficiently sampled. These arms are similar to the saturated arms in Thompson sampling, since the learner is confident about their sample mean reward estimates. However, different from Thompson sampling, in this work we have different sets of saturated arms for each type of context. Hence, our proposed algorithm exploits only when all arms are saturated for all types of contexts. Moreover Equation 3 enables us to control the number of explorations by D_{i,t}, thereby enabling us to achieve sublinear regret when reward observations are costly.

3) Due to the space limitations we have included the proof in the supplementary material. We will include a concise description of the proof and, importantly, of the key differences to standard proofs in the multi-armed bandit literature. A key difference is the identification of relevant types for each action, which requires estimating the variations of the expected arm rewards for different tuples of types to identify and rule out the set of types whose estimated rewards contradicts the Lipschitz similarity condition.

4) The introduction section will be revised and made more clear.

5) Thanks, all typos will be corrected.